# Differentiating Thyroid Follicular Adenoma from Follicular Carcinoma via G-Protein Coupled Receptor-Associated Sorting Protein 1 (GASP-1)

**DOI:** 10.3390/cancers15133404

**Published:** 2023-06-29

**Authors:** Yuan Rong, Cesar Torres-Luna, George Tuszynski, Richard Siderits, Frank N. Chang

**Affiliations:** 1Department of Pathology and Laboratory Medicine, Temple University School of Medicine, Philadelphia, PA 19140, USA; yuan.rong@tuhs.temple.edu; 2Halcyon Diagnostics, 1200 Corporate Blvd. Ste. 10C, Lancaster, PA 17601, USA; cesar.torres@lynthera.com (C.T.-L.); george.tuszynski@halcyonrx.com (G.T.); 3Department of Pathology and Laboratory Medicine, Robert Wood Johnson Medical School, Rutgers University, Piscataway, NJ 08854, USA; rick.siderits@dls.com

**Keywords:** thyroid cancer biomarker, thyroid nodule, immunohistochemistry, GASP-1, cancer prevention

## Abstract

**Simple Summary:**

Thyroid nodules are very common as one gets older. Fine-needle aspiration is the primary diagnostic methodology but difficulties in the diagnosis of follicular-derived thyroid lesions are well-known. The aim of this study was to assess the potential of G-protein coupled receptor-associated sorting protein 1 (GASP-1) as a valid thyroid cancer biomarker. By using anti-GASP-1 polyclonal antibodies in immunohistochemical (IHC) staining studies, we found that GASP-1 was significantly overexpressed in early-stage malignant thyroid neoplasms while it was minimally expressed in normal thyroid cells. We demonstrated that the level of GASP-1 expression can differentiate follicular adenoma from follicular carcinoma.

**Abstract:**

Follicular neoplasms are classified as benign or malignant depending on the presence or absence of capsular and/or vascular invasion. Due to incomplete capsular penetration or equivocal vascular invasion, the evaluation of these features can be challenging using histologic examination. In the current study, we analyzed the involvement of G-protein coupled receptor-associated sorting protein 1 (GASP-1) in the development and progression of thyroid neoplasms. Affinity-purified anti-GASP-1 polyclonal antibodies were used for routine immunohistochemistry (IHC) analysis. Thyroid tissue microarrays containing normal thyroid tissue, follicular adenoma, follicular carcinoma, papillary thyroid carcinoma, and anaplastic carcinoma were analyzed. We found that the level of GASP-1 expression can differentiate follicular adenoma from follicular carcinoma. When numerous cases were scored for GASP-1 expression by a board-certified pathologist, we found that GASP-1 expression is 7-fold higher in thyroid malignant neoplasms compared to normal thyroid tissue, and about 4-fold higher in follicular carcinoma compared to follicular adenoma. In follicular adenoma tissues, we observed the presence of many mini-glands that are enriched in GASP-1 and some mini-glands contain as few as three cells. GASP-1 IHC also possesses several advantages over the conventional H&E and can be used to identify early thyroid cancer and monitor cancer progression.

## 1. Introduction

The thyroid gland produces hormones that regulate the body’s metabolic rate, growth and development. It plays a role in controlling heart, muscle and digestive function, brain development, and bone maintenance [1]. The thyroid gland consists of two lobes, joined together by an intermediate structure, the isthmus. Each lobe contains many follicles, which are the structural and functional units of the thyroid gland [2]. The follicles are lined by follicular cells that rest on the basement membrane and have a cavity filled with a homogenous gelatinous material called the colloid. The space between the follicles is filled with connective tissue stroma, numerous capillaries, and lymphatics. It has been reported that the size of follicular cells varies depending on their metabolic activity [3]. When the follicles are in the resting (inactive) stage, the follicular cells are flattened. When the follicles are highly active, the follicular cells are mostly columnar. In a normal state of follicles during moderate activity, the cells are cuboidal, and the cavity is filled with a reasonable amount of colloid.

Thyroid cancer represents the most common endocrine malignancy, accounting for 3.4% of all cancers diagnosed annually [4,5,6]. The most common thyroid malignancy and the least aggressive is papillary thyroid carcinoma, with an incidence of 70–80% of all thyroid cancer cases. It is composed of multifocal papillary and follicular elements forming sites of carcinoma. Follicular carcinoma accounts for approximately 14% of thyroid cancers and is more aggressive than papillary thyroid carcinoma. Medullary thyroid carcinoma and anaplastic thyroid carcinoma are the other two types of thyroid cancer that represent less than 10% of the cases.

Thyroid nodules (TNs) are abnormal growths of thyroid cells that form lumps within the thyroid gland, which is located in the anterior neck region [7,8]. Radiologically, TNs are lesions within the thyroid gland that are distinct from the surrounding thyroid parenchyma. TNs are very common as one gets older, with TNs found in up to 50% of the population over the age of 60 [9,10]. It is reported that 90–95% of TNs are benign, with TNs being four times more common in women than in men, and their prevalence increasing with age and body mass index [11,12]. Other risk factors for thyroid cancer include ionizing radiation (e.g., from cancer treatments, occupational exposure), rapid nodule growth, hoarseness, and a family history of thyroid cancer or cancer syndromes [12,13,14].

The rise on the incidence of thyroid cancers is reported to be due to the widespread use of imaging studies, such as ultrasound, computed tomography, magnetic resonance imaging, and positron emission tomography [15,16,17]. The initial workup for any newly discovered TN includes the measurement of serum thyroid-stimulating hormone (TSH) level [18]. When thyroid hormone levels are low, the TSH rises responsively and vice versa; as a result, measuring the TSH level allows differentiation between functional and nonfunctional nodules [15]. When the initial workup hints at a nonfunctional nodule with suspicious sonographic features, a fine-needle aspiration (FNA) biopsy needs to be performed. FNA is the primary diagnostic methodology used for thyroid nodule evaluation and has proven to be of high value because nearly 70% of aspirates return benign, allowing for conservative management. However, difficulties in the diagnosis of follicular-derived thyroid lesions that are revealed by FNA cytology examination are well-known, and histologic evaluation of surgically resected follicular-derived lesions can be challenging as well. Follicular neoplasms are classified as benign or malignant depending on the presence or absence of capsular and/or vascular invasion [19,20]. Due to the presence of incomplete capsular penetration or equivocal vascular invasion, the evaluation of these features can be challenging on histologic examination. For this reason, 20–25% of thyroid biopsies are indeterminate, which means that a diagnosis between cancer and benign disease cannot be made by simply looking at the cells [21,22]. Patients with indeterminate thyroid biopsies are referred to surgery, resulting in many unnecessary surgeries for benign diseases.

We have previously identified that G-protein coupled receptor-associated sorting protein 1 (GASP-1) is a ubiquitous tumor marker and required for cancer progression and invasion [23,24,25,26,27,28]. One of our recent studies showed that GASP-1 promotes proliferation and invasion of the triple negative breast cancer cell line MDA-MB-231 [26]. Furthermore, it was demonstrated that a polyclonal antibody against the specific GASP-1 peptide EEASPEAVAGVGFESK inhibited growth and reduced the size of MDA-MB-231 cell colonies in soft agar by more than 90%. GASP-1 has not been previously reported as a thyroid cancer biomarker. In the current study, we analyzed the involvement of GASP-1 in the development and progression of thyroid neoplasms. For this work, we used thyroid tissue microarrays containing normal thyroid tissue, follicular adenoma, follicular carcinoma, papillary thyroid carcinoma, and anaplastic carcinoma.

## 2. Materials and Methods

**Thyroid tissue microarrays:** Thyroid tissue microarrays (TH201, TH641, TH8010a) purchased from US Biomax, Inc (Derwood, MD, USA) were used for routine H&E stain and GASP-1 immunohistochemistry. A total of 24 normal thyroid tissue specimens, 13 follicular adenoma specimens, 34 follicular carcinoma specimens, 59 papillary thyroid carcinoma specimens and 19 anaplastic (undifferentiated) carcinoma specimens were included in the study.

**Antibody production:** Anti-GASP-1 polyclonal antibodies against EEASPEAVAGVGFESK were produced by ABclonal Science, Inc (Woburn, MA, USA). Affinity-purified IgG on a column of the GASP-1 peptide EEASPEAVAGVGFESK were used for routine IHC analysis.

**Tissue staining:** Anti-GASP-1 polyclonal antibodies were used in immunohistochemical (IHC) staining to detect GASP-1 in formalin-fixed, paraffin-embedded (FFPE) tumor tissues. In brief, tissue staining was performed by Discovery Life Sciences (Newtown, PA, USA) according to standard procedures using optimized antibody concentration and antigen retrieval and the appropriate negative controls as described in another study [23]. Isotype non-immune IgG and immune sera adsorbed with GASP-1 peptide were used as controls and showed no staining.

**GASP-1 IHC scoring:** The intensity of GASP-1 expression for each specimen by immunohistochemistry was graded as 0, 1+, 2+ and 3+ using our previously published method [23]. Briefly, GASP-1 intensity is described as follows: 0 = No perceptible staining or no granules in cytoplasm; 1+: No granules observed at 20×, blush, or faintly perceptible powdery granule at 40× (weak); 2+: Fine-size granular pattern and easily perceptible intensity (moderate); 3+: Dense and dark cytoplasmic coarse granules (strong). The average GASP-1 expression score for a specimen in a particular group was calculated as the total GASP-1 score divided by the specimen numbers. The significance was calculated as the standard error of the mean. 

**Statistical analysis:** GraphPad Prism (version 8.0.0) was used to perform statistical analyses. The statistical significance of difference for two sample comparisons was determined by the unpaired Student t-test. The error bars represent the standard error of the mean. A *p* value of <0.05 was considered statistically significant.

## 3. Results


**GASP-1 expression in normal follicular cells**


We have previously reported that GASP-1 overexpression is required for breast cancer progression and invasion [23,24,25,26]. Breast cancer cells cannot grow or invade if GASP-1 expression is silenced. Thyroid follicular cells are known to be present in either resting (inactive) or metabolically active stage [3]. We decided to examine GASP-1 expression in different normal thyroid follicular cells using immunohistochemical staining with hematoxylin and GASP-1. Because GASP-1 is synthesized on the endoplasmic reticulum (ER), which is attached to the nucleus, the nucleus will initially appear as brown stain when GASP-1 is expressed there. Follicular cells from four normal individuals are presented in Figure 1. GASP-1 expression appears to correlate with the metabolic activity of the follicular cells. When the follicles are in the resting (inactive) stage, there is minimal or no expression of GASP-1 and the follicular cells are flat and squamous (Figure 1A,B). As follicular cells become metabolically active, there is an increase in GASP-1 expression and the size of nuclei increases (Figure 1C,D).


**Localized GASP-1 overexpression occurs in follicular adenoma**


Contrary to the expression pattern reported above in normal follicular cells, follicular adenoma appears to begin when GASP-1 overexpression occurs in specific areas of the follicles resulting in local cell overgrowth. GASP-1 expression in four different follicular adenomas is shown in Figure 2. While some follicular cells are metabolically inactive as judged by the lack of GASP-1 expression, other follicles are more actively expressing GASP-1. We can therefore identify the follicles (and the specific legion of the follicles) that are actively growing based on their GASP-1 expression. As follicular adenoma progressed, we also observed the appearance of many miniature-sized glands (called “mini-glands”). Some “mini-glands” contain as few as three cells, and they are highly enriched in GASP-1. We have expanded Figure 2D to show the mini-glands (see Figure 3D). To our knowledge, such GASP-1 mini-glands have not been previously reported. Although we do not know the significance and function of these GASP-1 mini-glands, they are associated with follicular adenoma and therefore could be used as one of the characteristics to identify follicular adenomas.


**GASP-1 IHC is superior to conventional H&E stain in identifying follicular adenoma**


H&E stain, firstly introduced in 1877 by chemist N. Wissozky, has been the most widely used stain in medical diagnosis and is often the gold standard method when a pathologist looks at a biopsy suspected of cancer. By using H&E, one can differentiate between the nuclear and cytoplasmic parts of a cell and the general layout and distribution of cells. Unlike eosin which stains many proteins, our GASP-1 IHC provides information regarding early development of benign tumors such as follicular adenoma.

A comparison of H&E and GASP-1 stains from the same follicular adenoma tissue shows that H&E has poor resolution and cannot detect early events in its progression (comparing Figure 3A and 3B). The presence of a large amount of thyroglobulin in the cavity of follicles causes a pink color to appear in large areas of the tissue, making it virtually impossible to detect the mini-glands observed in the GASP-1 IHC (comparing Figure 3C with 3D). Unlike the rather uniform GASP-1 expression in metabolically active follicles in normal thyroid tissue, the GASP-1 expression in follicular adenoma is uneven among the follicles. This uneven GASP-1 expression profile or the so-called “localized overgrowth” can be used to differentiate normal from follicular adenoma. As will be reported later, this localized overgrowth feature can also be used to differentiate follicular adenoma from follicular carcinoma.


**GASP-1 is highly overexpressed in follicular carcinoma**


The GASP-1 expression pattern of follicular carcinoma is vastly different from that of follicular adenoma. An example of GASP-1 expression in early stage (Stage I) follicular carcinoma is shown in Figure 4A. Unlike follicular adenoma described above, many follicular cells in follicular carcinoma are very active in expressing GASP-1. Overexpression of GASP-1 appears to be associated with an expansion of follicular cells resulting in a rather diffused appearance. Multiple layers of cancer cells with overexpressed GASP-1 are also observed. Compared to the resting follicular cells (see lower right corner in Figure 4A), the size of cells in follicular carcinoma is several times bigger. The overexpressed GASP-1 in the cytosol also starts to aggregate to form powdery granules. The interior of follicular cells is mostly filled with GASP-1 in early-stage follicular carcinoma. As follicular carcinoma progresses from early to late stages, continuous overexpression and aggregation of GASP-1 lead to bigger-size granules, which begin to attach to plasma membranes. GASP-1 is highly overexpressed in Stage 4 follicular carcinoma cells (Figure 4D).


**GASP-1 IHC has advantages over conventional H&E stain in assessing follicular carcinoma**


A comparison of H&E and GASP-1 IHC (Figure 5A,B) from the same follicular carcinoma tissue shows that H&E has poor resolution and makes it difficult to detect early events in cancer progression. GASP-1 IHC, on the other hand, clearly showed that in follicular carcinoma, large amounts of GASP-1 are produced and released into the cytosol producing a dark brown stain (Figure 5B). Figure 5C,D show expanded areas of both H&E and GASP-1 IHC. Our IHC also shows that the aggregation of overproduced GASP-1 in the cytosol forms large granules that begin to attach to the plasma membranes.


**GASP-1 is also highly overexpressed in papillary carcinoma**


The GASP-1 expression pattern of papillary carcinoma is similar to that of follicular carcinoma. This would be expected because both are cancerous conditions originating from follicular cells. Figure 6A shows the GASP-1 overexpression in Stage 1 papillary carcinoma. As cancer progresses, more GASP-1 is overexpressed, and the aggregation of cytosolic GASP-1 forms powdery and fine granules. Figure 6D shows the GASP-1 expression in anaplastic carcinoma which is a highly invasive cancer. Coarse GASP-1 granules are abundantly present, and many are attached to plasma membranes. Previously, we have observed such coarse granules and their attachment to plasma membranes in highly metastatic breast cancer [23].

In summary, GASP-1 overexpression is involved in the initiation of follicular adenoma, follicular carcinoma, and papillary carcinoma. While GASP-1 overexpression starts in limited areas of the follicles in follicular adenoma producing localized overgrowth, GASP-1 expression is highly increased in follicles of both follicular carcinoma and papillary carcinoma, resulting in widespread uncontrolled overgrowth normally found in many cancers.


**GASP-1 IHC score for differentiating follicular adenoma from follicular carcinoma**


A summary of GASP-1 expression scores for normal thyroid tissue, follicular adenoma, follicular carcinoma, papillary carcinoma, and anaplastic carcinoma is presented in Figure 7. GASP-1 expression is significantly higher in malignant thyroid neoplasms including follicular carcinoma, papillary carcinoma, and anaplastic carcinoma, compared to normal thyroid tissue and benign thyroid diseases (follicular adenoma) (*p* < 0.001). The average GASP-1 IHC score per specimen for normal individuals is 0.25 while the average score for follicular adenoma and follicular carcinoma is 0.46 and 1.75, respectively. Thus, GASP-1 expression is about 4-fold higher in follicular carcinoma when compared to that in follicular adenoma, and 7-fold higher when compared to the normal. This finding makes GASP-1 a potential immunohistochemical marker in differentiating benign from malignant follicular-derived thyroid nodules of FNA specimens. As expected, the more invasive anaplastic carcinoma has even higher GASP-1 IHC scores (2.15).

## 4. Discussion

The role of IHC in thyroid pathology includes multiple aspects of diagnosis as well as biomarkers that serve to provide information about prognosis, prediction, and genetic predisposition [29]. The current standard in the diagnosis of thyroid lesions is by histologic examination of routine H&E-stained sections. However, it is well known that identification of follicular lesions can be very difficult [30]. A common dilemma occurs when encapsulated tumors show a follicular growth pattern. The presence or absence of capsular and/or vascular invasion distinguishes benign from malignant follicular tumors, but substantiating this finding can be challenging due to incomplete capsular penetration or equivocal vascular invasion. Even though many thyroid diagnostic immunomarkers are expressed when carcinoma appears, none have been routinely adopted in cytological or histological diagnostic procedures due to high variation in their sensitivity and/or specificity. Some promising IHC markers for differential diagnosis of thyroid lesions include CD56, Hector Battifora mesothelial (HBME-1), galectin-3 (Gal-3) and CK19, but a combination of several of these markers must be used together in order to achieve high sensitivity and specificity [31]. A biomarker that can accurately assess the progression of thyroid carcinomas is highly desirable. Based on the presented results, our GASP-1 biomarker could fill this void.

In our previous work, we have demonstrated that GASP-1 overexpression is required for cancer progression and invasion [23,24,25,26]. Cancer cells fail to grow or invade when GASP-1 expression is silenced [23,26]. In the present study, we observed that the expression level of GASP-1 appears to correlate with the metabolic activity of normal thyroid follicular cells and as cancer progresses, more GASP-1 is overexpressed. The origin of follicular adenoma is currently unknown, and it is considered to have arisen differently from follicular carcinoma. Here, we showed for the first time that they have the same origin and that the extent of GASP-1 expression at an early stage is an important factor in determining whether it becomes follicular adenoma or follicular carcinoma. We observed that follicular adenoma begins with localized overexpression of GASP-1 resulting in overgrowth in those regions of follicles. Overgrowth of follicular cells produced a mixture of both normal (resting) and (single-layered) hyperplastic cells in follicular adenoma (see Figure 2). One surprising feature of follicular adenoma is that in all follicular adenoma tissues we observed the presence of many mini-glands that are enriched in GASP-1 and some mini-glands contain as few as three cells (see Figure 3D). We do not know the function of these “mini-glands” and hypothesize that due to being rich in GASP-1, which is a growth factor, they may promote the growth of supporting cells such as stromal cells and cause overgrowth of nodules in follicular adenoma. Further investigation involving more tissue samples will be required to substantiate this finding and the proposed hypothesis.

In contrast to the low level of expression in benign thyroid lesions, GASP-1 was significantly overexpressed in early-stage malignant thyroid neoplasms. We believe that follicular adenoma is a “local overgrowth” phenomenon, and the extent of overgrowth is related to the expression level of GASP-1. If so, GASP-1 expression level can be used to assess not only the beginning of follicular adenoma but also its progression.

A biomarker with high potential for both cancer diagnosis and prognosis should be able to differentiate between different cancer stages to avoid the progression to an advanced, invasive stage. Our results suggest that GASP-1 can be used as a potential immunohistochemical marker for distinguishing benign and malignant thyroid follicular-derived nodules for either fine needle aspiration specimens or resection specimens with difficulties in definite histological diagnostic criteria. Our GASP-1 IHC can also be used to assess the progression of papillary carcinomas. In our IHC score system, we found that GASP-1 expression is about 4-fold higher in follicular carcinoma compared to follicular adenoma. There is a 7-fold difference in staining score between normal thyroid and carcinomas, indicating that GASP-1 progression and overexpression can be differentiated across different stages of thyroid cancer (see Figure 4 and Figure 6). With the ability to differentiate follicular adenoma from early-stage follicular carcinoma based on the location of GASP-1 overexpression in the follicles, and the size and shape of follicular cells, GASP-1 IHC can be used to differentiate follicular adenoma from follicular carcinoma when conventional H&E IHC gives inconclusive results. Due to the readily availability of fine needle aspirate or tissue biopsy samples of many cancers, GASP-1 IHC is also well suited for assessing cancer progression. A future direction of the present work will be performing the GASP-1 staining in ambiguous cases of thyroid cancer and in FNA cytological specimens from medical or research institutions to confirm the real-world utility of GASP-1 as a thyroid cancer biomarker. In our future work, we will also conduct in vitro studies by using thyroid follicular cells (normal, immortalized, tumorigenic cells) and the manipulation of GASP-1 expression to unveil the causative role of GASP-1 in thyroid cell transformation.

## 5. Conclusions

GASP-1 overexpression is involved in the initiation of follicular adenoma and follicular carcinoma. The GASP-1 expression pattern can be used to differentiate normal thyroid tissue, follicular adenoma, and follicular carcinoma. While follicular adenoma starts with localized overexpression of GASP-1, a much higher level of GASP-1 expression covering larger areas of follicles is found in follicular carcinoma. GASP-1 level can also be used to assess the progression of follicular and papillary carcinomas. Immunohistochemical staining using GASP-1 possesses several advantages over the traditionally used H&E IHC. When there is ambiguity in disease assessment, such as differentiating follicular adenoma from follicular carcinoma in H&E, our GASP-1 IHC can be used to give a more clear-cut answer.

## Figures and Tables

**Figure 1 cancers-15-03404-f001:**
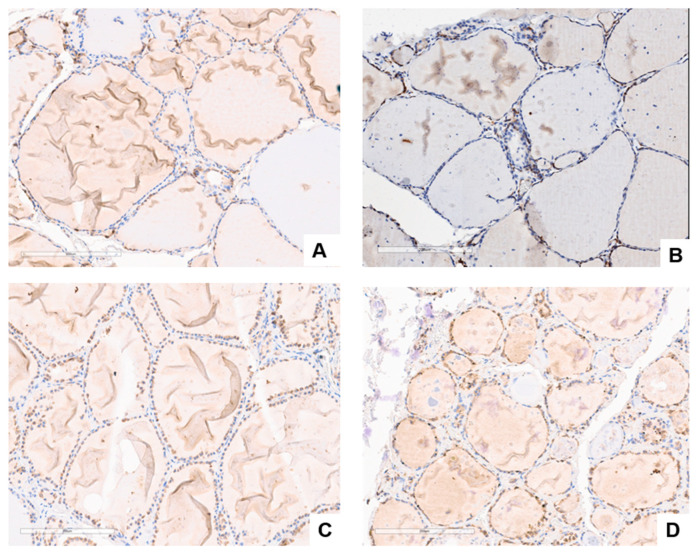
GASP-1 expression in normal thyroid tissues. (**A**–**D**) are normal thyroid tissues from different patients. Scale bar = 200 µm.

**Figure 2 cancers-15-03404-f002:**
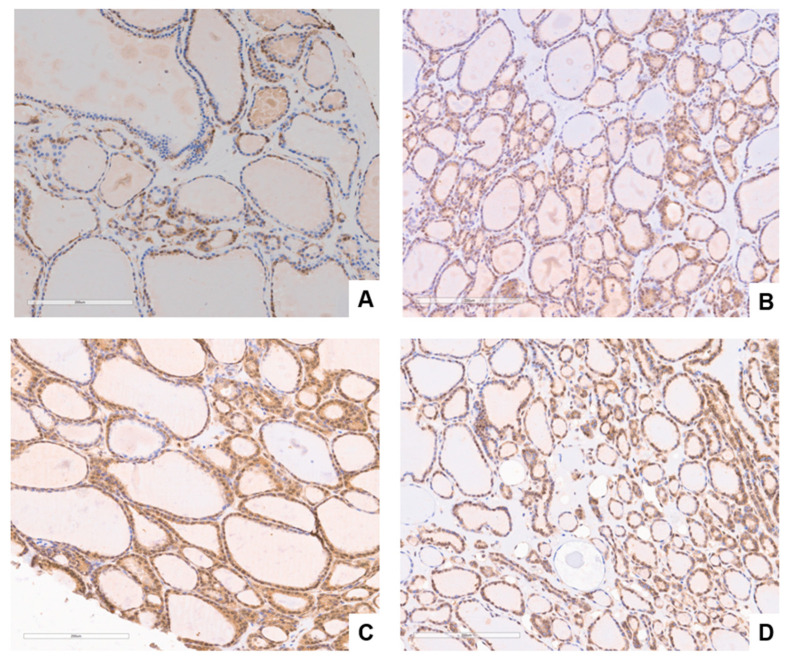
GASP-1 expression in follicular adenoma patients. Localized overexpression of GASP-1 in different follicular adenoma patients. (**A**–**D**) are tissues from different follicular adenoma patients. Scale bar = 200 µm.

**Figure 3 cancers-15-03404-f003:**
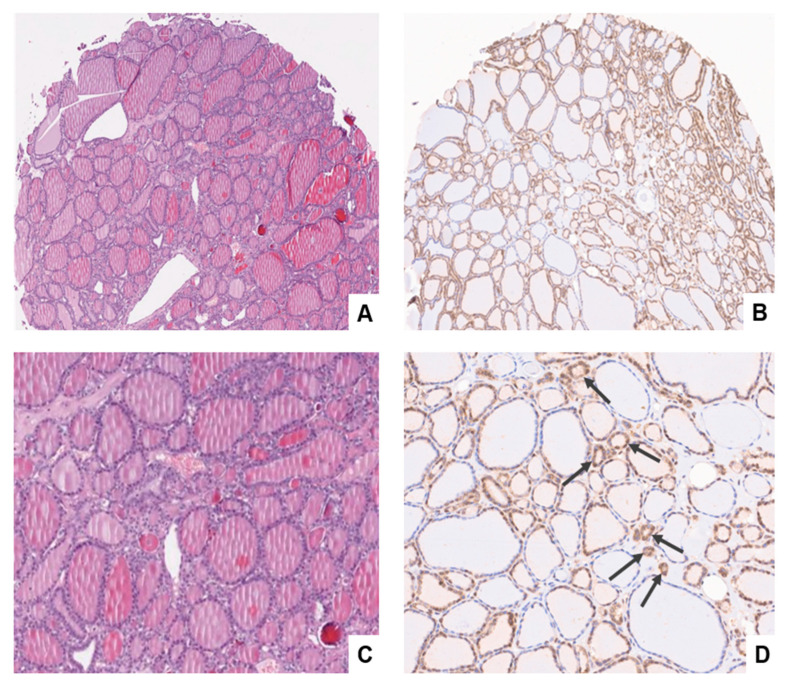
Comparison of H&E (**A,C**) and GASP-1 IHC (**B,D**) in same follicular adenoma tissue. (**C,D**) show expanded areas of both H&E and GASP-1 IHC. In Figure 3D, the arrows point to some of the mini-glands observed.

**Figure 4 cancers-15-03404-f004:**
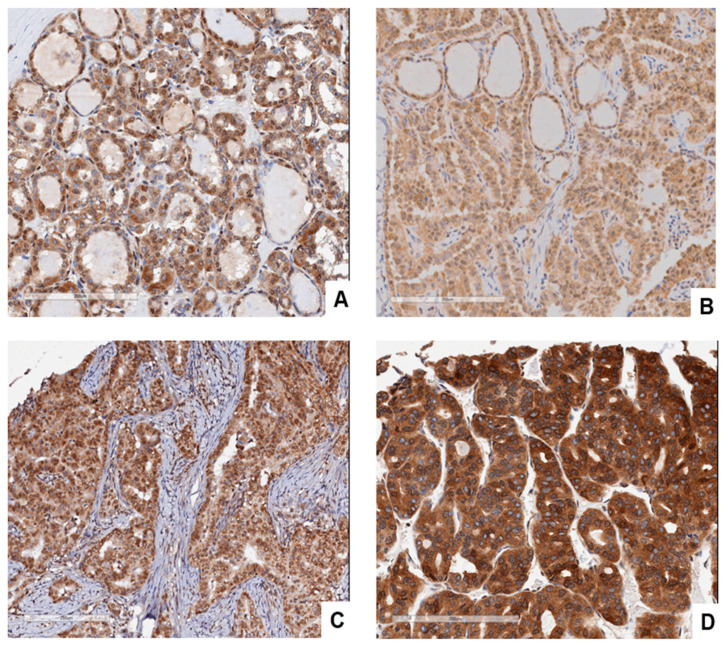
GASP-1 expression in different stages of follicular carcinoma patients. (**A**) (Stage 1), (**B**) (Stage 2), (**C**) (Stage 3), and (**D**) (Stage 4). Scale bar = 200 µm.

**Figure 5 cancers-15-03404-f005:**
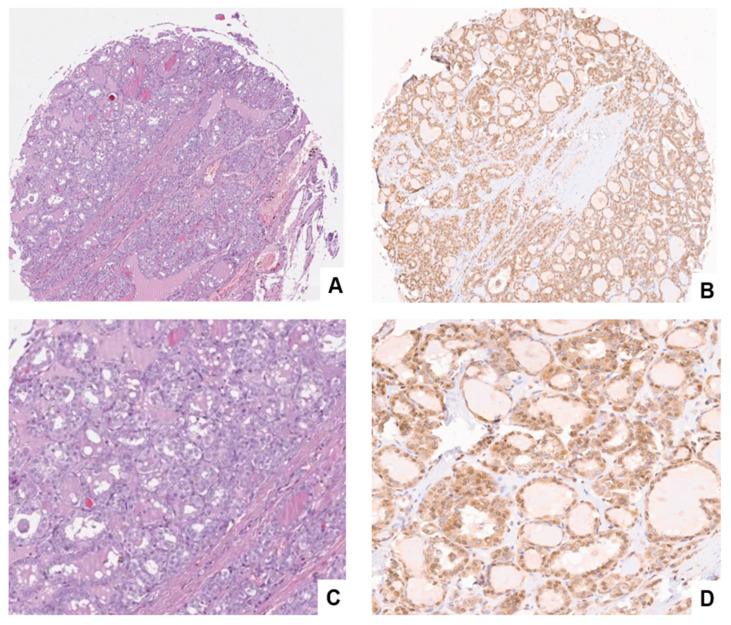
Comparison of H&E (**A,C**) and GASP-1 IHC (**B,D**) in Stage 1 follicular carcinoma tissue. (**C,D**) show expanded areas of both H&E and GASP-1 IHC.

**Figure 6 cancers-15-03404-f006:**
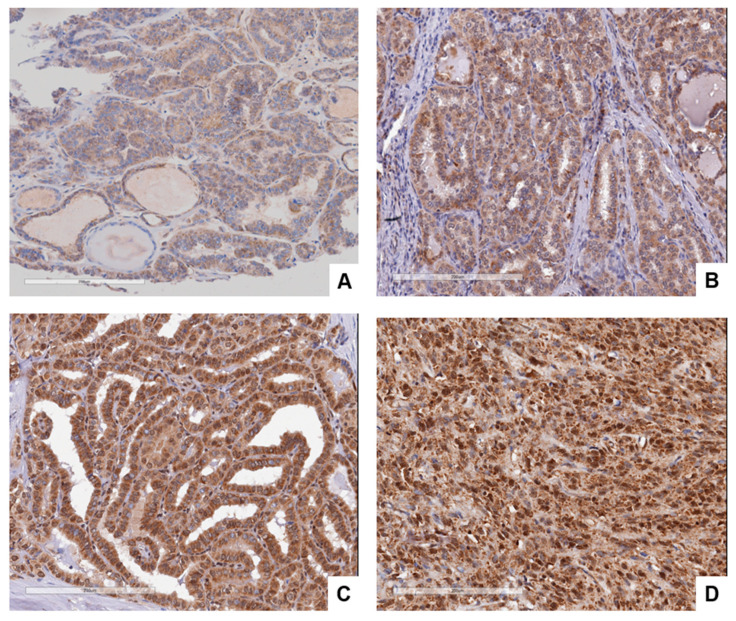
GASP-1 expression in different stages of papillary carcinoma patients and anaplastic carcinoma. (**A**) (Stage 1), (**B**) (Stage 2), (**C**) (Stage 3), and (**D**) (Anaplastic carcinoma). Scale bar = 200 µm.

**Figure 7 cancers-15-03404-f007:**
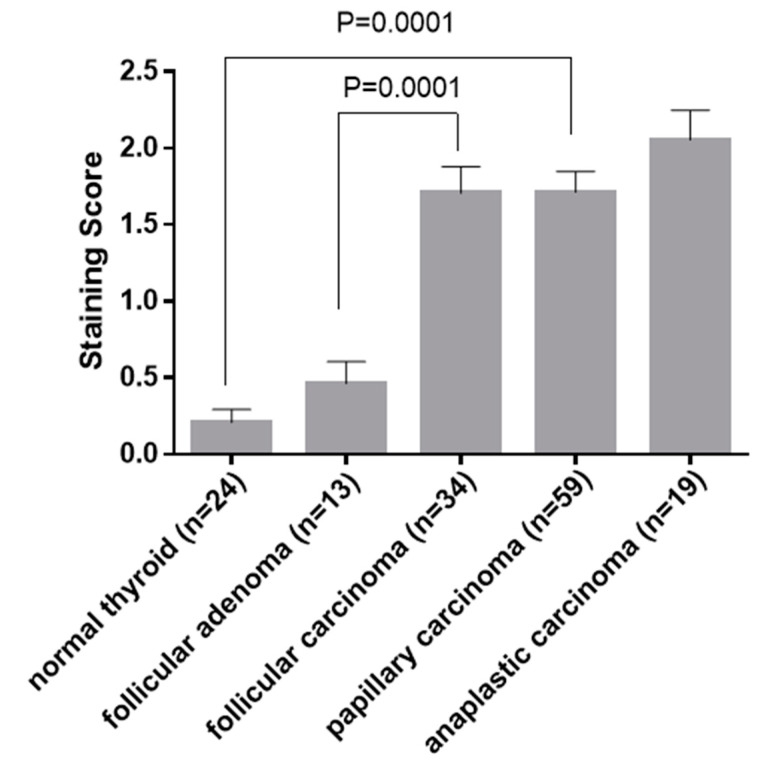
Average staining score of GASP-1 expression by immunohistochemistry. The error bars represent the standard error of the mean.

## Data Availability

The data generated in this study are available upon request from the corresponding author.

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
