# Peer review of "Differentiating Thyroid Follicular Adenoma from Follicular Carcinoma via G-Protein Coupled Receptor-Associated Sorting Protein 1 (GASP-1)"

_cancers, 2023, doi:10.3390/cancers15133404_

Round 1
Reviewer 1 Report
In the manuscript Rong et al. the authors used thyroid tissue microarrays to evaluate the use of GASP-1 as a possible thyroid cancer biomarker.
Although the manuscript is interesting I have some comments:
1) the authors claim that GASP-1 expression is increased in metabolically active normal follicular cells. to support this conclusion they show figure 1A and 1B as example of inactive follicular cells and figure 1C and D as example of metabolically active cells. However, it is not clear how they discriminate between the two. it would be useful if they could clarify this point especially because all figure 1 panels are supposed to be from normal thyroid tissue. is there any other marker that could be used to confirm the difference in metabolic state?
2) In figure 2 the authors show the presence of areas where GASP-1 expression is increased that they refer to as "mini glands". It would help if they could point at them in figure 2 panels.
3) the authors claim that GASP-1 is highly overexpressed in follicular carcinoma leading to uncontrolled cell growth. It is not clear to me how they reach the conclusion that GASP-1 causes cell growth and is not just an association. it would be different if they had performed overexpression experiments in a cell line like Nthy-ori 3-1 and observed increased proliferation.
4) the raw data for figure 7 should be included as supplementary.
Author Response
- The authors claim that GASP-1 expression is increased in metabolically active normal follicular cells. To support this conclusion they show figure 1A and 1B as example of inactive follicular cells and figure 1C and D as example of metabolically active cells. However, it is not clear how they discriminate between the two. it would be useful if they could clarify this point especially because all figure 1 panels are supposed to be from normal thyroid tissue. is there any other marker that could be used to confirm the difference in metabolic state?
Response: We have revised the mentioned statement (lines 149-155) regarding GASP-1 expression in normal follicular cells.
- In figure 2 the authors show the presence of areas where GASP-1 expression is increased that they refer to as "mini glands". It would help if they could point at them in figure 2 panels.
Response: To point at the mini-glands, we have included “arrows” pointing out some of these mini-glands in Figure 3D (which is a higher magnification figure of Figure 2D).
- The authors claim that GASP-1 is highly overexpressed in follicular carcinoma leading to uncontrolled cell growth. It is not clear to me how they reach the conclusion that GASP-1 causes cell growth and is not just an association. it would be different if they had performed overexpression experiments in a cell line like Nthy-ori 3-1 and observed increased proliferation.
Response: We have revised the statements in the manuscript regarding GASP-1 highly overexpression leading to uncontrolled cell growth (lines 205-206, lines 234-236, lines 320-324).
- The raw data for figure 7 should be included as supplementary.
Response: Authors believe that this figure should be included in the main text as it is providing a summary of the average GASP-1 IHC scores; by this way it is possible to quantitatively compare GASP-1 expression among different stages of thyroid cancer
Reviewer 2 Report
In the article “Differentiating thyroid follicular adenoma from follicular carcinoma via G-protein coupled receptor-associated sorting protein 1 (GASP-1)” Rong et al. analyzed GASP1 protein levels in normal as well as in several thyroid cancers by immunohistochemistry, correlating its expression and intracellular localization with cancer progression.
One issue is to understand whether the authors are proposing GASP1 as a new biomarker to identify follicular cell transformation and/or a new driver for thyroid cell transformation. As demonstrated for breast cancer in prevoius works, to propose GASP1 as a new driver “required for the initiation and progression of thyroid cancers” more biological studies should be included in this report. In vitro studies by using thyroid follicular cells (normal, immortalized, tumorigenic cells) and manipulation of GASP1 expression (overexpression and downregulation) are necessary to unveil the causative (if it is) role of GASP1 in thyroid cell transformation. In addition, other systems (in vitro as well as in vivo) will be necessary to distinguish GASP1 role in promoting or sustaining (later stage) thyroid cancer. Thus, the authors should revise sentences that are not supported by experiments in the paper (28-29;220-201 lines). Lines 205-207 “Continuous overexpression of GASP-1 and their release into the cytoplasm caused large expansion of follicular cells producing a rather diffused appearance” how the authors demonstrate that the releasing of GASP CAUSE a large expansion of follicular cells?
-In all the immunohistochemical analysis a higher magnification should be included to appreciate intracellular GASP localization (fig 1;fig2, fig 3,fig 4…)
-In fig 3 the author showed how increased GASP1 levels in thyroid tissues may histologically identify follicular adenomas. Can the author stain two consecutive slides derived from the same sample, one with H&E and the other with GASP1? This may help to show the different structures that can be identify by GASP IHC compared to H&E. The miniglands are just formed by few (possibly proliferating) cells. Can the author show in figures the area presenting miniglands? Can we observe Miniglands also by Hemathoxilin by identifying aggregation of more than 3/4 follicular cells? Can we stain Miniglands by KI67 IHC?
- in fig 4 can the author include H&E of samples stained for GASP? This may help to follow the stages of thyroid follicular transformation and the association of GASP expression. Do the follicular carcinomas included in the figure display the same molecular signature?
-Are there any correlation between GASP1 expression and specific molecular thyroid cancer signature? For example, are there any difference between GASP1 expression and Papillary thyroid cancer harboring B-Raf mutations or RET rearrangements; are there any difference between GASP1 expression and follicular thyroid cancers carrying RAS or PI3K mutations?
Minor points:
-The statistical analysis to prove the significance of differences of more than 2 samples (as occur by comparing several type pf thyroid cancers, e.g papillary, follicular, anaplastic cancers) can be also obtained by using different approaches such as anova test.
- Did the author evaluate GASP IHC in thyroid nodules of FNA specimens or in paraffin embedded thyroid tissues?
-Lines 237-238 “This would be expected because both are cancerous conditions originating from follicular cells” Do the author evaluate GASP expression in medullary thyroid speciments? How is the expression of GASP in parafollicular cells? Do the author evaluate GASP1 expression in hyperthyroid patient?
Author Response
- The authors should revise sentences that are not supported by experiments in the paper (28-29;220-201 lines). Lines 205-207 “Continuous overexpression of GASP-1 and their release into the cytoplasm caused large expansion of follicular cells producing a rather diffused appearance” how the authors demonstrate that the releasing of GASP CAUSE a large expansion of follicular cells?
Response: We have revised lines 28-29, 212-213, 226-227. Sentences have been either revised or eliminated.
- In all the immunohistochemical analysis a higher magnification should be included to appreciate intracellular GASP localization (fig 1;fig2, fig 3,fig 4…)
Response: We have already provided higher magnification of GASP-1 expression in both Figures 3 and 5. In Fig.3, we selected an image from Figure 2 to compare GASP-1 expression with H&E staining. Similarly, in Fig. 5 we selected an image from Figure 4 to compare GASP-1 expression with H&E staining.
- In fig 3 the author showed how increased GASP1 levels in thyroid tissues may histologically identify follicular adenomas. Can the author stain two consecutive slides derived from the same sample, one with H&E and the other with GASP1? This may help to show the different structures that can be identify by GASP IHC compared to H&E. The miniglands are just formed by few (possibly proliferating) cells. Can the author show in figures the area presenting miniglands? Can we observe Miniglands also by Hemathoxilin by identifying aggregation of more than 3/4 follicular cells? Can we stain Miniglands by KI67 IHC?
Response: We agree that serial sections stained with GASP-1 followed by H&E would highlight the unique cancer signature provided by GASP-1 expression. The H&E image provided by the TMA provider for Fig. 3 is not a serial section and does not provide additional information as to the cancer diagnostic potential of GASP-1 expression. We have drawn arrows in Figure 3D (which is a higher magnification figure of Figure 2D) to point some of the mini-glands in the image. It will be possible to observe mini-glands by hemathoxilin or by KI67 IHC and this will be part of the future work that the authors want to perform for this project.
- In fig 4 can the author include H&E of samples stained for GASP? This may help to follow the stages of thyroid follicular transformation and the association of GASP expression. Do the follicular carcinomas included in the figure display the same molecular signature?
Response: Authors haven’t done H&E staining of samples stained for GASP-1 in Figure 4 but it will be part of the future work as we agree with the reviewer that this will help to follow the stages of thyroid follicular transformation and its association with GASP expression. We don’t have the molecular signature information of the follicular carcinomas evaluated in Figure 4 and how this could impact staining but in our next studies we will be evaluating the impact of molecular profiling on GASP-1 overexpression.
- Are there any correlation between GASP1 expression and specific molecular thyroid cancer signature? For example, are there any difference between GASP1 expression and Papillary thyroid cancer harboring B-Raf mutations or RET rearrangements; are there any difference between GASP1 expression and follicular thyroid cancers carrying RAS or PI3K mutations?
Response: As mentioned in our previous response, we haven’t yet performed studies to correlate GASP-1 expression with specific molecular thyroid cancer signature. However, it is of deep interest to the authors to analyze the impact of molecular signature on the GASP-1 expression and this is going to be conducted in our future experiments.
- Minor point: The statistical analysis to prove the significance of differences of more than 2 samples (as occur by comparing several type pf thyroid cancers, e.g papillary, follicular, anaplastic cancers) can be also obtained by using different approaches such as anova test.
Response: In our study we were evaluating the statistical significance of difference for two sample comparisons (i.e., follicular adenoma vs. normal tissue; or follicular carcinoma vs. follicular adenoma) using unpaired Student t test. As a result, the authors didn’t consider necessary to use anova test to determine statistically significant differences among means of multiple groups.
- Minor point: Did the author evaluate GASP IHC in thyroid nodules of FNA specimens or in paraffin embedded thyroid tissues?
Response: GASP-1 IHC was evaluated in paraffin embedded thyroid tissues
- Minor point: Lines 237-238 “This would be expected because both are cancerous conditions originating from follicular cells” Do the author evaluate GASP expression in medullary thyroid speciments? How is the expression of GASP in parafollicular cells? Do the author evaluate GASP1 expression in hyperthyroid patient?
Response: GASP-1 expression hasn’t been evaluated in medullary thyroid specimens, parafollicular cells, or hyperthyroid patients. This work will be conducted in our futures studies.
Reviewer 3 Report
In the manuscript titled “Differentiating thyroid follicular adenoma from follicular carcinoma via G-protein coupled receptor-associated sorting protein 1 (GASP-1)”, the authors have analyzed normal thyroid tissues, follicular cell derived thyroid tumors (benign and malignant) as well as anaplastic carcinomas with regards to GASP-1 protein expression by immunohistochemistry. They conclude that when there is ambiguity in disease assessment such as differentiating follicular adenoma from follicular carcinoma in H&E, GASP-1 IHC can be used to give a more clear-cut answer. Although the study was conducted on a relatively small sample size, it is a well-designed study and the manuscript is well written with clear methodology and results. However, a few concerns need to be addressed by the authors, before the manuscript can be accepted.
1. Introduction section (Page 2): The first paragraph of the Introduction section can be removed entirely since it describes the basic anatomy and physiology of the thyroid gland, which is not relevant to the study.
2. The authors state that GASP-1 protein expression can be used to differentiate between follicular adenoma and follicular carcinoma based on their results. However, have they performed the staining in ambiguous cases from their institute to reiterate this finding. If not, this needs to be mentioned as a possible limitation or future direction of the study. It could be an interesting study to analyze the utility of GASP-1 staining in real-world specimens (rather than readily available TMAs).
3. Another real-world utility of this staining could be on FNAC specimens, which could provide the clinicians a clear idea with regards to further management based on the staining results. However, this too would require a future study analyzing the sensitivity, specificity and diagnostic accuracy of GASP-1 IHC staining in differentiating follicular adenoma from follicular carcinoma on cytology specimens compared to histopathological specimens. This too can be mentioned as a possible limitation or future direction of the study.
Minor grammatical errors need to be corrected.
Author Response
- Introduction section (Page 2): The first paragraph of the Introduction section can be removed entirely since it describes the basic anatomy and physiology of the thyroid gland, which is not relevant to the study.
Response: The authors consider that the first paragraph regarding the basic anatomy and physiology of the thyroid gland is relevant to provide both the “lay” audience and scientists that are non-specialized in thyroid cancer with basic understanding of the thyroid gland's anatomy and physiology.
- The authors state that GASP-1 protein expression can be used to differentiate between follicular adenoma and follicular carcinoma based on their results. However, have they performed the staining in ambiguous cases from their institute to reiterate this finding. If not, this needs to be mentioned as a possible limitation or future direction of the study. It could be an interesting study to analyze the utility of GASP-1 staining in real-world specimens (rather than readily available TMAs).
Response: The authors agree that performing the staining in ambiguous cases from medical institutions will be important to reiterate the finding. This has been mentioned in the Discussion as a future direction of the study.
- Another real-world utility of this staining could be on FNAC specimens, which could provide the clinicians a clear idea with regards to further management based on the staining results. However, this too would require a future study analyzing the sensitivity, specificity and diagnostic accuracy of GASP-1 IHC staining in differentiating follicular adenoma from follicular carcinoma on cytology specimens compared to histopathological specimens. This too can be mentioned as a possible limitation or future direction of the study.
Response: The authors agree that staining on FNAC specimens is a future direction that needs to be stated in the study. This has also been mentioned in the Discussion section as a future direction.
Round 2
Reviewer 2 Report
thanks for the reply